# Enhanced Timeliness and Co-Administration of Meningitis B Vaccination in Children: Impact of Funding in Valencian Community, Spain

**DOI:** 10.3390/vaccines12060623

**Published:** 2024-06-05

**Authors:** Juan Juaneda, Pablo Estrella-Porter, Carolina Blanco-Calvo, Alejandro Orrico-Sánchez, José Antonio Lluch-Rodrigo, Eliseo Pastor-Villalba

**Affiliations:** 1Directorate-General for Public Health, Conselleria de Sanitat, 46010 Valencia, Spain; estrella_pab@gva.es (P.E.-P.); blanco_carcal@gva.es (C.B.-C.); jall@comv.es (J.A.L.-R.); pastor_eli@gva.es (E.P.-V.); 2Preventive Medicine and Public Health, Hospital La Fe, 46026 Valencia, Spain; 3Preventive Medicine, Hospital Clínico Universitario, 46010 Valencia, Spain; 4Preventive Medicine, Hospital Universitario Doctor Peset, 46017 Valencia, Spain; 5Vaccines Research Unit, Fundación para el Fomento de la Investigación Sanitaria y Biomédica de la Comunitat Valenciana (FISABIO), 46017 Valencia, Spain; alejandro.orrico@fisabio.es; 6CIBER of Epidemiology and Public Health (CIBERESP), Instituto de Salud Carlos III, 28029 Madrid, Spain; 7Universidad Católica de Valencia San Vicente Mártir, 46001 Valencia, Spain

**Keywords:** immunisation schedule, *Neisseria meningitidis* serogroup B, vaccination, vaccination coverage, program evaluation

## Abstract

Public funding of vaccines may enhance vaccination rates, co-administration, and timeliness. The impacts of including the serogroup B meningococcus vaccine (MenB) into the national immunisation schedule on vaccination rates, co-administration rates, and timeliness were assessed using a population-based pre-funding (2022) and post-funding (2023) study design. MenB vaccination rates improved after funding and were in line with previously funded vaccines. Co-administration rates also increased significantly. Timely administration increased, protecting children at an early age. Public funding has a positive impact on vaccine accessibility and early protection. Consistent population characteristics highlight the role of funding.

## 1. Introduction

Invasive meningococcal disease (IMD) is a major public health concern caused by the bacterium *Neisseria meningitidis* (also known as meningococcus). In Spain, between the 2012/13 and 2021/22 seasons, the main serogroup causing IMD in each season was MenB, with a total of 1295 cases. It has the highest number of cases in children under five and the highest incidence rate in children under one, compared with other serotypes. Last season, it accounted for 48.1% of total cases (n = 108) and 73.2% of typeable cases (n = 71) [1,2].

The lifetime societal cost per case of IMD has been estimated to be as high as USD 319,896, including both direct healthcare costs and wider societal impacts such as productivity losses and long-term care requirements [3]. A study conducted in Spain showed that the effectiveness against IMD caused by any serogroup of partial (one dose) or complete (two doses) vaccination with the four-component, protein-based meningococcal serogroup B (4CMenB) vaccine was 54.0% (95% CI, 18 to 74) and 76.0% (95% CI, 57 to 87), respectively. Against meningococcal serogroup B disease, the effectiveness was 64.0% (95% CI, 41 to 78) and 71.0% (95% CI, 45 to 85), respectively [4].

In the Valencian Community, vaccination against IMD caused by serogroup B meningococcus (MenB) is recommended and funded for all individuals born from January 2023 onwards. The current vaccination strategy involves an investment of EUR 60 per dose, with three doses per child. The cost to the annual budget of the Directorate-General for Public Health is approximately EUR 6.5 million for an average of 36,000 births per year [5].

The 4CMenB vaccine was incorporated into the national immunisation schedule following a 2 + 1 dosing pattern, with two doses administered at 2 and 4 months of age and a booster dose at 12 months [2]. The first dose at 2 months is co-administered with the hexavalent vaccine (Hex-V) and the anti-pneumococcal conjugated 13-valent vaccine (PCV13). The second dose at 4 months is co-administered with Hex-V, PCV13, and serogroup C anti-meningococcal vaccine (MenC). The booster dose at 12 months is co-administered with MenC and measles–mumps–rubella vaccine [6].

Prior to 2023, MenB vaccines were self-funded by families on the recommendation of paediatricians, reaching a moderate coverage rate of around 66.4% for the 2018–2021 cohorts of the Valencian Community [7]. Furthermore, studies in the region have shown that children who had not received the MenB vaccine were at higher risk of social exclusion [7,8]. In a societal preference study for MenB vaccination in Spain, it was revealed that cost was a primary determinant in parental decision-making, with household income emerging as the most influential variable [9]. According to the World Health Organization, sustainable financing of immunisation is essential to achieve universal coverage [10].

Public funding of vaccine programmes can positively impact vaccination coverage and timely administration. As previously shown in Spain, public funding of anti-pneumococcal vaccination had a significant impact on coverage rates and timeliness [11].

Promoting co-administration with other vaccines is an efficient and cost-saving strategy that has been shown to increase immunisation rates in children and adults [12]. A study conducted in England between 2008 and 2018 by Bauwens et al. found that only 64% of vaccines scheduled for co-administration in children were co-administered as recommended. Timely vaccine administration is critical to achieving recommended vaccine co-administration. In particular, a delay in the administration of at least one vaccine significantly reduced the likelihood of achieving recommended vaccine co-administration [13].

The aim of this study was to compare vaccination rates, timely administration, and the co-administration of the MenB vaccine between the pre-funding (2022) and post-funding (2023) periods.

## 2. Material and Methods

### 2.1. Study Design and Population

A pre–post observational descriptive study was carried out. All children born between 1 January and 31 July in 2022 and 2023 in the Valencian Community were included. Children whose mothers lived outside the Valencian Community were excluded from the study.

### 2.2. Data Sources

#### 2.2.1. Neonatal Screening Registry

All births were extracted from the Neonatal Screening Registry. The registry includes all children who have undergone the heel-prick test as part of the Neonatal Screening for Congenital Diseases. This screening is a population-based measure applied to all live births in the Valencian Community [14]. In the year 2021, the total coverage of the registry was 99.9% of the 35,810 births in the Valencian Community [15]. It serves as the primary data source for this study, providing information on births in the Valencian Community, including data on the birth centre and demographic information on the mothers of the children.

#### 2.2.2. Vaccine Information System

Vaccination data were extracted from the regional Vaccine Information System by October of each year. The Vaccine Information System of the Valencian Community has stored all information on vaccinations since the year 2000, with comprehensive data available since 2005 [16,17]. It records the vaccines administered in both public and private centres, including data on vaccine type, manufacturer, batch number, date and place of administration, associated risk group, and reported adverse reactions, covering both funded and unfunded vaccines.

### 2.3. Study Variables

Population characteristics were described and compared between the 2022 and 2023 cohorts, including sex, birth weight, gestational week at birth, mother’s age, and mother’s origin countries/regions.

The vaccinations analysed were the MenB vaccine and is schedule-indicated counterparts at 2 and 4 months of age. The vaccines given according to the schedule were the first doses of Hex-V and PCV13 at 2 months of age and the second doses of Hex-V and PCV13, along with the first dose of MenC, at 4 months of age. If the serogroup ACWY anti-meningococcal vaccine (MenACWY) was given instead of MenC, it was considered as a valid dose.

The difference in the time of administration between years was calculated for each dose.

### 2.4. Outcomes

#### 2.4.1. Vaccination Rate

The vaccination rate was calculated by dividing the number of children vaccinated with the *n*th dose by the number of children born in the period *m*, where *n* was the number of the dose (first or second dose) and *m* referred to the study period (January to July 2022 or January to July 2023).

#### 2.4.2. Co-Administration Rate

Co-administration was defined as the administration of the MenB vaccine on the same day as the other vaccines given according to the schedule. The MenB vaccine could be co-administered with Hex-V and PCV13 at 2 months of age and with Hex-V, PCV13, and MenC/MenACWY at 4 months of age.

An indicator for the co-administration of the MenB vaccine with each of the other vaccines was created, as well as an indicator for the co-administration of the MenB vaccine with all other scheduled vaccines. This last indicator showed whether all the vaccines recommended in the schedule, plus the MenB vaccine, were administered in the same vaccination session, making the most of vaccination opportunities.

#### 2.4.3. Timely Administration

Timely administration was considered to have been achieved if the vaccination was given within the month of age for which the vaccine is recommended by public health authorities’ schedules (for MenB, first dose at 2 months and second dose at 4 months of age) [2,6]. For example, for a child born on 1 January, the first and second doses were considered timely if they were given between 1 March and 31 March or 1 May and 31 May, respectively.

### 2.5. Statistical Analysis

Previously defined study variables and outcomes were compared between the two cohorts.

The Mann–Whitney U test was used to assess the statistical significance of the continuous variables due to the non-normal distribution of the variables. Pearson’s chi-squared test was used for categorical variables. A *p*-value of <0.05 was considered statistically significant.

All statistical analyses were performed using R 4.3.0 and RStudio 2023.06.2 build 561.

## 3. Results

Between January and July 2022, 20,054 children were born, compared to 19,909 born between January and July 2023. Table 1 shows the characteristics of the two cohorts studied and their statistical comparison. The number of births by month and year is shown in Figure 1.

There were no significant differences in terms of sex, birth weight, and gestational weeks at birth between the 2022 and 2023 cohorts. Statistically significant differences were found for maternal age, but the absolute differences were small. The most relevant difference was found for the mothers’ origin WHO region, where Spanish mothers represented 69.7% in 2022 and 61.2% in 2023 (Table 1).

### 3.1. Vaccination Rates (VR)

Of the total births in 2022 (n = 20,054), 13,392 (VR = 66.8%) individuals received one dose of MenB vaccine, and 12,020 (VR = 60.0%) individuals received two doses. These rates were substantially lower than those for Hex-V, PCV13, and MenC/MenACWY, which ranged from 94.4% to 96.1% for the first doses and from 94.1% to 94.3% for the second doses. In 2023 (n = 19,909), the number of vaccinated individuals increased to 18,990 (VR = 95.3%) for the first dose and 18,315 (VR = 92.0%) for the second dose, reaching similar levels compared to Hex-V, PCV13, and MenC/MenACWY. Detailed vaccination rates are shown in Table 2.

### 3.2. Co-Administration

Of those vaccinated with the first dose of MenB, 12.6% were co-administered with Hex-V and PCV13 in 2022, while in 2023 the co-administration accounted for 89.8%. For the second dose, 10.3% were co-administered with Hex-V, PCV13, and MenC/MenACWY in 2022. This rate also increased in 2023, reaching a rate of 81.6%. Detailed co-administration rates by vaccine are shown in Figure 2.

### 3.3. Timely Administration

In the cohort of children born in 2022, 14.7% and 11.1% of the first and second doses, respectively, were administered on time. For children born in 2023, these rates increased to 88.4% and 81.8% for the first and second doses, respectively. The time of administration of the first and second doses decreased significantly between the study years (−4.4 and −5.0 median weeks, respectively, *p* < 0.0001). The distribution of administration times by year by dose is shown in Figure 3.

When all doses and years were compared, there were no significant differences by sex, maternal age, country of origin, birth weight, or gestational week at birth.

## 4. Discussion

The introduction of public funding for the MenB vaccine in the Valencian Community and its subsequent evaluation have provided valuable insights into the importance of health policies for achieving public health objectives.

In 2022, prior to vaccine funding, MenB vaccine coverage was notably lower than other vaccines already included in public funding programmes, namely Hex-V, PCV13, and MenC. Following the introduction of funding for MenB in 2023, coverage reached levels similar to other vaccines, reinforcing the idea that public funding serves as a key determinant of vaccination coverage.

The significant increase in co-administration rates for the MenB vaccine with Hex-V, PCV13, and MenC/MenACWY in 2023, compared to 2022, shows that public funding of the MenB vaccine facilitated the integration of this vaccine into the established immunisation schedule. In addition, the study showed a remarkable improvement in the timely administration of the MenB vaccine post-funding. Timely administration of vaccines is key to providing early protection to infants against vaccine-preventable diseases. These findings add to the evidence that public funding of vaccines significantly improves the efficiency of vaccine delivery, in terms of coverage, timeliness, and co-administration.

The results of this study are consistent with the previous Madrid study [11], which found than public funding not only increased vaccination rates but also improves the timeliness of vaccine administration.

Most studies of vaccine coverage aim to assess vaccine effectiveness. However, very few evaluate the impact of funding on vaccine uptake and more investment should be made in this type of research. Howard et al. assessed the impact of funded influenza vaccine programmes in Australia by comparing vaccine uptake before and after the introduction of funded vaccines for children aged 6 months to less than 5 years and showed that government-funded vaccines can lead to an almost fivefold increase in vaccine uptake in the target age group. They also found that funded vaccines for young children may encourage caregivers to also vaccinate themselves and their older children [18]. Most importantly, De Oliveira Bernard et al. highlighted that the patients attending practices in the most affluent areas had the highest vaccination coverage in all years and they benefited most from this Australian government-funded influenza programme [19].

It is worth noting that there were no important or significant differences in population characteristics such as sex, maternal age, birth weight, or gestational weeks at birth between the two cohorts. There were significant differences in the region of origin of the mother.

This points to the role of funding as the primary driver behind increased vaccination rates and the timely administration observed in this study.

Nonetheless, this research is not devoid of limitations. First of all, there was a decrease of almost 9 percentage points in the population of Spanish origin when comparing the 2022 and 2023 cohorts, mainly at the expense of an increase in unknown origin, for which no data are available. Other researchers have shown that socioeconomic factors are associated with vaccine uptake. Ijalba-Martínez et al. found a significant association between vaccination status and national origin [20]. Taking this into account, if the entire unknown data belonged to foreigners, the increase in coverage, co-administration, and timely administration could be overestimated, considering that these groups could benefit more from the implementation of a funded vaccine schedule. Conversely, if they were Spanish, there would be no differences in the distribution of this variable between both cohorts. Secondly, the study was limited to the Valencian Community, so the extrapolation of these results to other contexts may be limited. Finally, the study did not assess potential barriers to vaccination or reasons for non-compliance with the recommended vaccination schedule, which could provide additional insights for improving vaccination strategies. Further research is needed, in particular, to assess whether socioeconomic factors affecting access to vaccination persist after the cost of vaccination is covered. Potential areas for investigation include the impact of migration status (such as language barriers and difficulties in accessing the health system), vulnerability status, family size, income-related factors, and health insurance coverage.

## 5. Conclusions

This study highlights the positive impact of public funding on MenB vaccination rates, co-administration of vaccines, and timely administration. The findings suggest that public health policies that focus on financial accessibility can make a significant contribution to increasing equitable access to the MenB vaccine and protecting children as early as possible, according to the technical schedules. Future research should explore other potential strategies to improve vaccination rates and the efficiency of vaccine delivery in different contexts.

## Figures and Tables

**Figure 1 vaccines-12-00623-f001:**
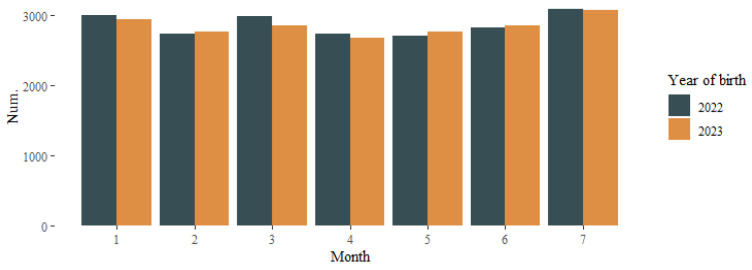
Number of births by month according to year of birth.

**Figure 2 vaccines-12-00623-f002:**
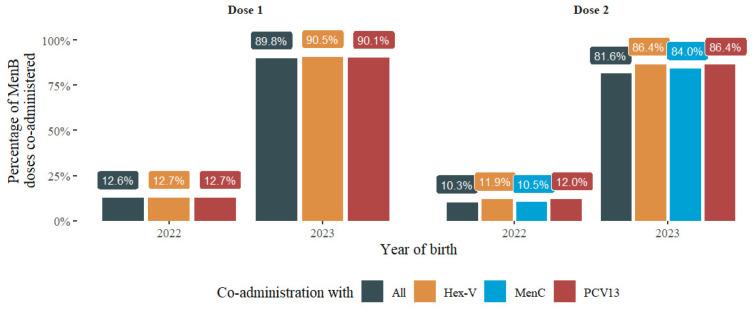
Percentage of MenB doses co-administered with vaccines indicated by schedule according to dose and year of birth.

**Figure 3 vaccines-12-00623-f003:**
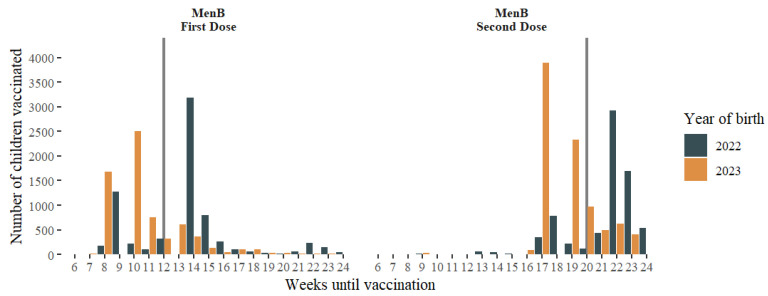
Distribution of the time of administration (weeks until vaccination) by year of birth, according to MenB dose (vertical lines show the maximum date for each dose to be considered as timely administered).

**Table 1 vaccines-12-00623-t001:** Population characteristics by age of birth.

Characteristic	Overall n = 39,930 ^1^	2022 n = 20,031 ^1^	2023 n = 19,899 ^1^	*p*-Value
**Sex**				0.9 ^2^
Men	20,493 (51.3%)	10,288 (51.4%)	10,205 (51.3%)	
Women	19,436 (48.7%)	9742 (48.6%)	9694 (48.7%)	
**Weight at birth**	3245 (2930, 3550)	3240 (2920, 3550)	3250 (2930, 3550)	0.11 ^3^
**Gestational weeks at birth**	39.00 (38.0, 40.0)	39.00 (38.0, 40.0)	39.00 (38.00, 40.00)	0.12 ^3^
**Maternal age**	33.0 (29.0, 36.0)	33.0 (29.0, 37.0)	33.0 (28.0, 36.0)	<0.001 ^3^
**Spanish mother**	26,139 (65.5%)	13,966 (69.7%)	12,173 (61.2%)	<0.001 ^2^
**Mother’s origin WHO Region** *				<0.001 ^2^
Spain	26,139 (65.5%)	13,966 (69.7%)	12,173 (61.2%)	
Africa	730 (1.8%)	373 (1.9%)	357 (1.8%)	
Americas	4656 (11.7%)	2154 (10.8%)	2502 (12.6%)	
Eastern Mediterranean	2845 (7.1%)	1468 (7.3%)	1377 (6.9%)	
Europe	3239 (8.1%)	1667 (8.3%)	1572 (7.9%)	
South East Asia	165 (0.4%)	71 (0.4%)	94 (0.5%)	
Western Pacific	219 (0.5%)	119 (0.6%)	100 (0.5%)	
Unknown	1937 (4.9%)	213 (1.1%)	1724 (8.7%)	

^1^ n (%); Median (IQR). ^2^ Pearson’s Chi-squared test. ^3^ Mann–Whitney U test. * WHO Region: World Health Organization Region (Spain separated).

**Table 2 vaccines-12-00623-t002:** Vaccination rates of vaccines analysed, by dose and year.

Year (n)	2022 n = 20,054		2023 n = 19,909	
Vaccine	Dose 1	Dose 2	Dose 1	Dose 2
MenB	66.8%	60.0%	95.3%	92.0%
Hex-V	96.1%	94.3%	95.8%	93.6%
MenC/MenACWY ^1^	94.4%	-	93.6%	-
PCV13	96.1%	94.1%	95.8%	93.4%

^1^ First dose is given at the same time as the second doses of PCV13, Hex-V, and MenB. Second dose rate is not calculated as it is administered at 12 months of age (outside the study period). Empty cells are indicated by “-”.

## Data Availability

Data are unavailable due to privacy or ethical restrictions as mandated by Ley Orgánica 7/2021, de 26 de mayo, de protección de datos personales tratados para fines de prevención, detección, investigación y enjuiciamiento de infracciones penales y de ejecución de sanciones penales (Organic Law 7/2021 of 26 May, on the protection of personal data processed for the purposes of crime prevention, detection, investigation, and prosecution, as well as the enforcement of criminal sanctions) and Regulation (EU) 2016/679 of the European Parliament and of the Council of 27 April 2016, concerning the protection of natural persons with regard to the processing of personal data and on the free movement of such data, and repealing Directive 95/46/EC (General Data Protection Regulation).

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
