# Peer review of "Enhanced Timeliness and Co-Administration of Meningitis B Vaccination in Children: Impact of Funding in Valencian Community, Spain"

_vaccines, 2024, doi:10.3390/vaccines12060623_

Round 1

Reviewer 1 Report

Comments and Suggestions for Authors

The paper describes the impact of reimbursability of meningococcal B vaccination in the Valencian pediatric population in terms of vaccination coverage, efficiency, and timeliness.

Here are my suggestions:

- Please add some references regarding the burden of MenB in pediatrics or the general population in Spain.

- Please add "observational" at line 63 between "pre-post" and "descriptive study"

- Please, consider including at least one decimal number in all the percentages and a single decimal number for continuous variables of Table 1

- Regarding "Mother's origin country" consider grouping some groups (es. using the WHO regions) and evaluate the differences using Fisher's exact test (although it isn't necessary)

As a final comment, I enjoyed the discussion. In particular, the section on study limitations is well described. Consider expanding it by referencing other studies regarding the socioeconomic factors highlighted in line 219.

Author Response

Dear Reviewer,

Thank you for your valuable feedback on our manuscript. We have made the following revisions based on your suggestions:

  1. Added references to the burden of MenB in children and the general population in Spain, as suggested, in the Introduction.

  2. Included the term "observational" in the description of the study in the Study Design and Population section, as recommended.

  3. Formatted percentages and numbers as suggested.

  4. Grouped the country of origin of the mothers according to the WHO regions, with Spain excluded, and evaluated the differences using the chi-squared test. NAs were explicitly added to the variable "Mother's country of origin WHO region" in Table 1 for better understanding.

  5. Expanded the references on socioeconomic factors in the Discussion section.

Thank you once again for your insightful comments.

Best regards,

Reviewer 2 Report

Comments and Suggestions for Authors

Nice study with clearly stated results.  Would suggest just adding some discussion either in introduction or Discussion about the cost burden of meningococcal disease and how effective the vaccine is against invasive disease over a life

Comments on the Quality of English Language

Line 48 could be rewritten "Public funding of vaccine programs can positively impact vaccination coverage and timely administration."

Author Response

Dear Reviewer,

Thank you for your positive feedback on our manuscript. We have made the following revisions based on your suggestions:

  1. Added references to the cost burden of meningococcal disease and the effectiveness of the vaccine against invasive disease in the Introduction.

  2. Undertook the suggested minor editing of the English language throughout the manuscript, including the revision of line 48 to "Public funding of vaccine programs can positively impact vaccination coverage and timely administration."

Thank you once again for your insightful comments.

Best regards,

Reviewer 3 Report

Comments and Suggestions for Authors

The authors point out that the cost of MenB vaccination in Spain is the primary consideration for the parental decision-making. The public funding on vaccines can facilitate the vaccination coverage and timely administration as proved by another vaccine (anti-pneumococcal vaccination). It is an encouragement to see the year of birth of 2023 has a much higher percentage compared with that of the year of birth of 2023 since the timely vaccination is a prophylactic protection. In addition, to their knowledge, the authors did not find any significantly differences by sex, maternal age, country of origin, birth weight, and gestational week at birth in response to all doses and years’ comparison.

I have some recommendations and comments as below:

1. May I know that how much funding for MenB vaccine has been awarded in 2023? It is important to know how much money is needed to have such a large improvement.

2. The authors finally point out the limitations of no investigating other reasons that can contribute to the low rate of MenB vaccination. I am wondering if the authors can list 2-3 items that can be potentially investigated.

Author Response

Dear Reviewer,

Thank you for your valuable feedback on our manuscript. We have made the following revisions based on your suggestions:

  1. Included information on the funding allocated to the MenB vaccine in 2023 in the Introduction, highlighting the importance of understanding the financial investment required to achieve significant improvements.

  2. Suggested potential areas for further research in the Discussion section, specifically identifying 2-3 factors that may contribute to the low uptake of MenB vaccination.

Thank you once again for your insightful comments.

Best regards,